# Shared Recurrent Memory Improves Multi-agent Pathfinding

**Alsu Sagirova**[1,2]   **Yuri Kuratov**[1,2]   **Mikhail Burtsev**[3]

[1]AIRI, Moscow, Russia   [2]Neural Networks and Deep Learning Lab, MIPT, Dolgoprudny, Russia
[3]London Institute for Mathematical Sciences, London, UK
{alsu.sagirova, yurii.kuratov}@phystech.edu, mb@lims.ac.uk

## Abstract

Multi-agent systems are gaining attention in AI research due to their ability to solve complex problems in a distributed manner, but coordinating multiple agents remains challenging. Inspired by the global workspace theory, we introduce the Shared Recurrent Memory Transformer (SRMT).[1] SRMT extends memory transformers to multi-agent settings by pooling and globally broadcasting individual working memories, enabling agents to implicitly exchange information and coordinate behavior. We evaluate SRMT on the Partially Observable Multi-Agent Path Finding in a bottleneck navigation task requiring agents to pass through a narrow corridor. SRMT consistently outperforms a range of reinforcement learning baselines, especially under challenging reward structures with sparse feedback. Our experiments also demonstrate that SRMT generalizes effectively to environments with significantly longer corridors than those seen during training, highlighting its scalability and robustness. These results suggest that incorporating shared memory structures into transformer-based architectures can enhance coordination in decentralized multi-agent systems.

## 1   Introduction

Multi-agent systems have significant potential to solve complex problems through distributed intelligence and collaboration. However, coordinating the interactions between multiple agents remains challenging, often requiring sophisticated communication protocols and decision-making mechanisms. We propose a novel approach to address this challenge by introducing a *shared memory* as a global workspace for agents to coordinate behavior. The global workspace theory [Baars, 1988] suggests that in the brain, there are independent functional modules that can cooperate by broadcasting information through a global workspace. Inspired by this theory, we consider the agents in MAPF as independent modules with shared memory and propose a *Shared Recurrent Memory Transformer* (SRMT) as a mechanism for exchanging information to improve coordination and avoid deadlocks. SRMT extends memory transformers [Burtsev et al., 2020, Bulatov et al., 2022, Yang et al., 2022, Cherepanov et al., 2024] to multi-agent settings by pooling and globally broadcasting individual working memories.

In this study, we test the shared memory approach on Partially Observable Multi-agent Path Finding (PO-MAPF) [Stern et al., 2019], where each agent has to reach its goal while observing the state of the environment, including locations and actions of the other agents and/or static obstacles, only locally. Multi-agent pathfinding can be considered in the *decentralized* manner, where each agent independently collects rewards and makes decisions on its actions. There are many attempts to solve

---

[1]https://github.com/Aloriosa/srmt

this problem: in robotics [Van den Berg et al., 2008, Zhu et al., 2022], in machine and reinforcement learning field [Damani et al., 2021, Ma et al., 2021b, Wang et al., 2023, Sartoretti et al., 2019, Riviere et al., 2020]. Such methods often involve manual reward-shaping and external demonstrations. Also, several works utilize the communication between agents to solve decentralized MAPF [Ma et al., 2021a, Li et al., 2022, Wang et al., 2023]. Yet still, the resulting solutions are prone to deadlocks and poorly generalizable to the maps not used for training. In this work, we compare SRMT to a range of RL baselines and show that it consistently outperforms them in the bottleneck navigation task.

## 2 Shared Recurrent Memory Transformer

In SRMT, the input sequence for each agent at each time step is constructed by combining three key components: the agent's personal memory vector; the historical sequence of the agent's observations from the past $h = 8$ time steps; and the current step's observation. This sequence undergoes a full self-attention mechanism. Next, the output of the self-attention is passed through a cross-attention layer between current hidden representations and shared memory. The shared memory consists of a globally accessible, ordered sequence of all agents' memory vectors for the current time step. This interaction between personal and shared memory enables each agent to incorporate global context into its decision-making process. The resulting output is then passed through a memory head, which updates the agent's personal memory vector for the next time step. Simultaneously, the output is also fed into the decoder part of the policy model, which generates the agent's action (Fig. 1A).

We use POGEMA [Skrynnik et al., 2024a] framework for the experiments. In POGEMA the two-dimensional environment is represented as a grid composed of obstacles and free cells. At each time step each agent can perform two types of actions: moving to an adjacent cell or remaining at their current position. Agents have limited observational capabilities, perceiving other agents only within a local $5 \times 5$ area centered on their current position. The episode ends when the predefined time step, episode length, is reached. The episode can also end before this time step if certain conditions are met, i.e. all agents reach their goal locations.

As an initial test of our approach, we selected a simple two-agent coordination task where the agents must navigate through a narrow passage. The environment consists of two rooms connected by a one-cell-wide corridor, as illustrated in Fig. 1B. Each agent starts in one room, and their goal is located in the opposite room, requiring both agents to pass through the narrow corridor to complete the task. To train the model, we utilized 16 maps with corridor lengths uniformly selected between 3 and 30 cells.

To evaluate the performance, we utilize three metrics: *Cooperative Success Rate (CSR)* – a binary measure indicating whether all agents reached their goals before the episode's end; *Individual Success Rate (ISR)* – the fraction of the agents that achieved their goals during the episode; and *Sum-of-Costs (SoC)* – the total number of time steps taken by all agents to reach their respective goals (the lower the value the better). The policy function is approximated with a deep neural network (see Appendix Fig. 4). The agents are assumed to be homogeneous, so the policy is shared between agents during training. To benchmark the effectiveness of SRMT, we compared it against several baselines: recurrent memory transformer without memory sharing (RMT), transformer without memory (Attention), direct connection of spatial encoder to actor-critic action decoder (Empty), GRU RNN (RNN). Among these baselines, RMT, Attention core and Empty core methods can be considered as ablations of SRMT architecture. We also trained and evaluated MAMBA [Egorov and Shpilman, 2022] and QPLEX [Wang et al., 2020] as baselines. Additional details regarding the training procedure can be found in Appendix A.1.

## 3 Results and Discussion

We first evaluated SRMT against the baseline models using three variations of the reward function: *Directional*, *Moving Negative*, and *Sparse*.[2] In the Directional setting, the agent was rewarded for reaching the goal and for every step that brought it closer to the goal. In Moving Negative movements are slightly penalized for minimizing the path to the goal but no information about the goal location is provided. In contrast, the Sparse reward function only gives a reward when the agent successfully moves into the goal cell, offering no intermediate rewards.

---

[2]See Appendix A for details of rewards and extra results.

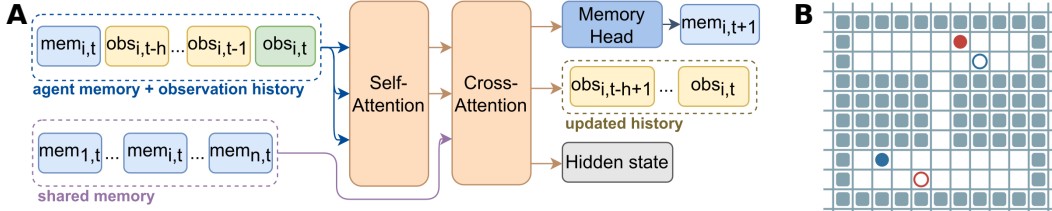

Figure 1: **Model and task overview.** A. Shared Recurrent Memory Transformer (SRMT) pools recurrent memories $mem_{i,t}$ of individual agents at a moment $t$ and provides global access to them via cross-attention. B. In the *Bottleneck Task* two agents start in rooms opposite to their goals and should coordinate passing the corridor. Agents are shown as solid colored circles; their goals are empty circles with the same colored border.

The comparison of evaluation scores, shown in Figure 2, demonstrates that SRMT successfully solves the task under considered reward functions. In particular, SRMT consistently performed well, even in the challenging Moving Negative and Sparse reward scenarios where other methods struggled. Considering the Sparse reward, the difference between SRMT and RMT scores is statistically significant (Wilcoxon signed-rank test, $p < 0.001$). The ability to coordinate across agents via shared memory proved critical, especially when feedback from the environment was minimal.

Among the non-memory sharing methods, RMT achieved the best performance, outperforming both the GRU-based RNN and the vanilla transformer models. This indicates that the combination of self-attention with recurrence in RMT offers distinct advantages in multi-agent coordination, particularly in tasks that require sequential decision-making over time. However, without the shared memory, RMT's performance in the Sparse reward setting was still limited compared to SRMT, highlighting the importance of global information exchange.

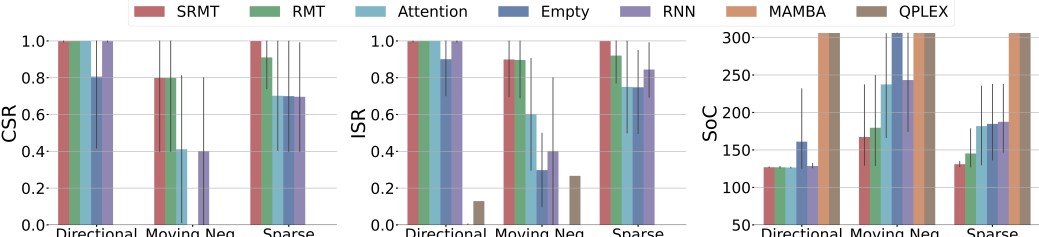

Figure 2: **SRMT effectively solves the Bottleneck Task with different reward functions.** Trained with Directional (positive when moved towards a goal and achieved it) reward, all the methods except Empty core policy, MAMBA, and QPLEX solve the task. For the case with the negative reward for movement and no directional reward (Moving Negative) memory-based methods (SRMT and RMT) demonstrate a clear advantage over memory-less (Attention, Empty, RNN). With the Sparse (only on-goal) reward, SRMT maintains the score while other methods drop. Error bars indicate 95% confidence intervals. For CSR and ISR higher values are better, for SoC - the lower the better.

To further assess the generalization capabilities of the trained policies, we evaluated them on bottleneck environments with corridor lengths significantly larger than those used during training, ranging from 5 to 1000 cells. The evaluation results[3] for the Sparse and Moving Negative reward functions are shown in Figure 3.

The results indicate that trained with the Sparse reward function, the SRMT-based policy consistently outperforms the baseline methods regarding both the individual success rate (ISR) and the total time spent in the environment (SoC). Remarkably, SRMT scales effectively up to corridor lengths of 400 cells without any significant performance degradation. For corridor lengths beyond 400 steps, the Cooperative Success Rate (CSR) of SRMT drops from 1.0 to 0.8, which still surpasses the other models, except for RMT. RMT shows the highest stability across all tested corridor lengths,

---

[3]For these evaluations, we adjusted the episode length to $2 \cdot corridor\_len + 100$ to ensure that both agents had sufficient time to reach their goals within each episode.

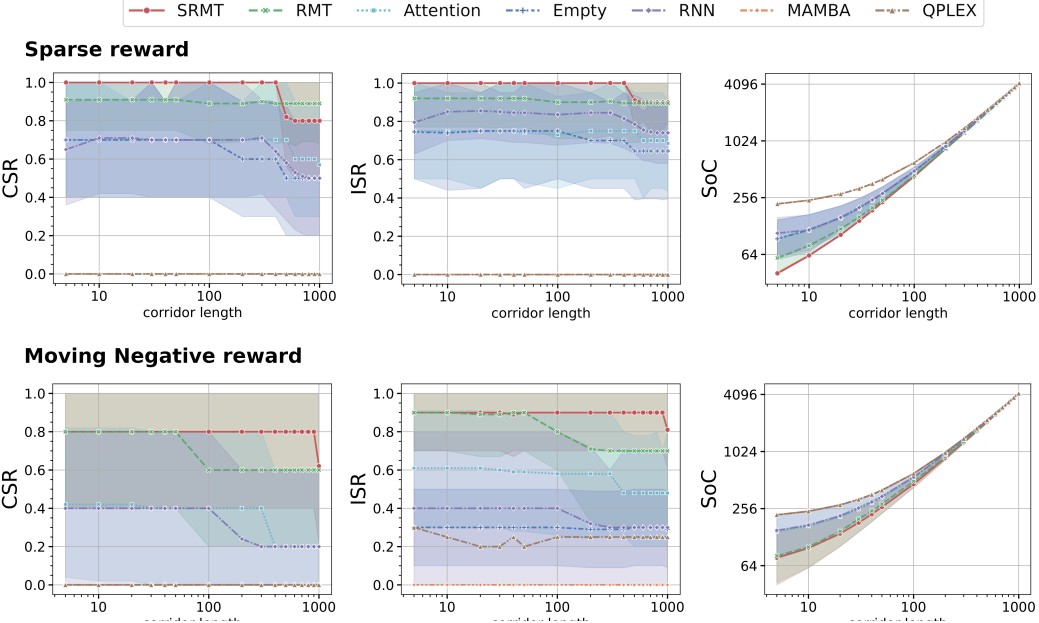

Figure 3: **SRMT agents generalize on corridor lengths up to 1000.** After training on corridor sizes from 3 to 30 cells all methods were evaluated on longer passages up to 1000. All models show good scaling till 100 while memory-based transformers were able to scale beyond that. For the Sparse reward, SRMT leads up to 400 and then drops below RMT for collective performance. For the Moving Negative reward, SRMT shows the top-1 performance on all three metrics. The shaded area indicates 95% confidence intervals.

maintaining a score of 0.9 for CSR across the board. For the Moving Negative reward function, resulting policies with SRMT show the top-1 scores for any corridor length up to 1000 (see Fig. 3). Evaluation scores for the other reward functions and analysis of memory representations can be found in Appendix A.

## 4 Conclusion

In this paper, we introduced the Shared Recurrent Memory Transformer (SRMT) as a novel architecture for enhancing coordination in multi-agent systems. SRMT enables agents to exchange information implicitly and coordinate actions without explicit communication protocols. Our experimental results on the bottleneck navigation task demonstrate that SRMT consistently outperforms baseline models, especially in challenging scenarios with sparse rewards and extended corridor lengths. The shared memory mechanism allows agents to generalize their learned policies to environments with significantly longer corridors than those seen during training, demonstrating the scalability and robustness of our approach. These findings highlight the potential of incorporating shared memory structures in transformer-based architectures for multi-agent reinforcement learning.

## Limitations

As in the majority of research related to Multi-Agent Path Finding (MAPF), in this work we assume that the agents have flawless localization and mapping abilities. Our primary focus is on the decision-making aspect of the problem. We also consider that the agents execute actions accurately and that their moves are synchronized. Additionally, we treat obstacles as fixed elements of the environment.

Finally, it's important to note that our approach, like other prominent learnable methods designed for (PO)-MAPF – such as PRIMAL [Sartoretti et al., 2019], PRIMAL2 [Damani et al., 2021], DHC [Ma et al., 2021a], and PICO [Li et al., 2022] – does not offer theoretical guarantees that agents will reach their destinations. However, extensive experimental evidence, both from our work and from the

referenced studies, demonstrates that these learnable methods are practically powerful and scalable solutions for complex MAPF problems.

## Acknowledgments and Disclosure of Funding

This work was partially supported by a grant for research centers, provided by the Analytical Center for the Government of the Russian Federation in accordance with the subsidy agreement (agreement identifier 000000D730324P540002) and the agreement with the Moscow Institute of Physics and Technology dated November 1, 2021 No. 70-2021-00138.

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
