# A    Appendix

## A.1    Training details

The environments were created with POGEMA [Skrynnik et al., 2024a] framework. The Sample Factory codebase [Petrenko et al., 2020] was used for policy model training.

The policy neural network is presented in Figure 4. We adopt the Skrynnik et al. [2024b] approach for configuring the model and input-output data pipelines. The model input is the agent's local observation tensor of shape $3 \times m \times m$ tensor, where $m$ is the observation range. The channels of the tensor encode the static obstacle locations combined with the current path, the other agents, and their targets.

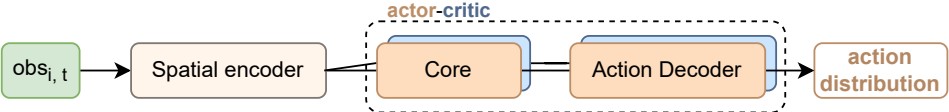

Figure 4: **Learnable policy architecture.**. The actor-critic consists of the core subnetwork and action decoder.

The spatial encoder is a ResNet [He et al., 2016] with an additional Multi-Layer Perceptron (MLP) in the output layer. The Action Decoder and the Critic Head are Dense layers. As the core sub-network, we consider the GRU architecture [Chung et al., 2014] or the attention block implemented in the Huggingface GPT-2[4] model.

We apply no advanced heuristics or methods for path planning to test the impact of memory addition. Our path-planning strategy is simple: each agent aims to follow the shortest path to the goal at each time step. If according to the planned movements the agents are to collide, their final decision will be to retain their current positions until the next step. We hypothesize that a shared memory mechanism will help to solve such bottleneck problems.

Training parameters for all tested methods are listed in Table 1. A single Tesla P100 was used for training policy models for approximately 1 hour each. The results for models trained with Sparse and Dense reward functions were averaged over 10 runs with different random seeds. The results of training policies with Directional and Directional Negative rewards were averaged over 5 runs with different random seeds as they showed less variation during training. Each run evaluation was first averaged over 10 different evaluation procedure random seeds. We carried out the grid search for the SRMT training entropy coefficient (range $[0.00001, 0.0003]$) and learning rate (range $[0.01, 0.05]$).

## A.2    Evaluation scores

In this section, we provide the evaluation results for Dense, Directional, and Directional Negative reward functions. The Figures 5, 6, 7 have superior or comparable performance compared to the baselines.

## A.3    Memory Analysis

We also explored the relations between the SRMT agents' memory representations and the spatial distances between agents on the map. Fig. 8 shows that SRMT distances between memory representations are aligned with distances between agents for different corridor lengths. Starting the episode, the agents move closer to each other quickly, and the respective cosine distances decrease significantly. Then, agents face each other in the environment (marked with a triangle on Fig. 8) and move in the same direction, keeping the spatial distance between them constant. Next, after the moment when one of the agents reaches its goal and disappears from the environment (marked with a star), the other agent moves away to reach the goal. This part of the episode is depicted as increasing memory distance at the end of the episode.

---

[4]https://huggingface.co/docs/transformers/model_doc/gpt2

Table 1: Training hyperparameters.

| Parameter | Value |
| --- | --- |
| Optimizer | Adam |
| Learning rate | 0.00013 |
| LR Scheduler | Adaptive KL |
| $\gamma$ (discount factor) | 0.9716 |
| Recurrence rollout | 8 |
| Clip ratio | 0.2 |
| Batch size | 16384 |
| Optimization epochs | 1 |
| Entropy coefficient | 0.0156 |
| Value loss coefficient | 0.5 |
| $GAE_\lambda$ | 0.95 |
| MLP hidden size | 16 |
| ResNet residual blocks | 1 |
| ResNet filters | 8 |
| Attention hidden size | 16 |
| Attention heads | 4 |
| GRU hidden size | 16 |
| Activation function | ReLU |
| Network Initialization | orthogonal |
| Rollout workers | 4 |
| Envs per worker | 4 |
| Training steps | $2 \times 10^7$ |
| Episode length | 512 |
| Observation patch | $5 \times 5$ |
| Number of agents | 2 |

Table 2: Tested reward functions. We list the reward values for achieving the goal, moving on the path toward the goal, or taking other actions (moving not in the direction of the goal and staying in the same position).

| Type | On goal | Move towards goal | Else |
| --- | --- | --- | --- |
| Directional | +1 | +0.005 | 0 |
| Sparse | +1 | 0 | 0 |
| Dense | +1 | -0.01 | -0.01 |
| Directional Negative | +1 | -0.005 | -0.01 |
| Moving Negative | +1 | -0.01 | -0.01 for moving, -0.005 for holding |

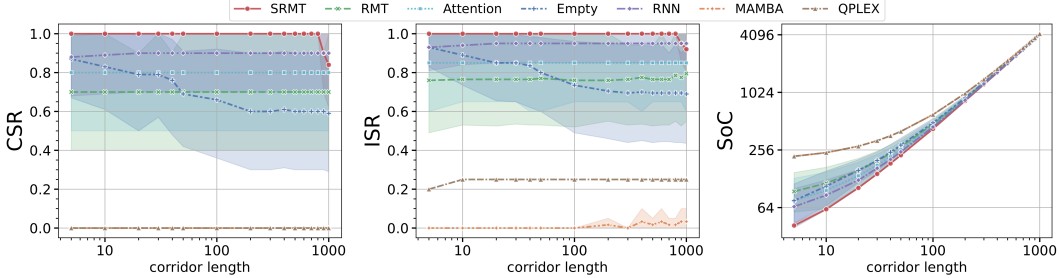

Figure 5: Trained with **Dense** reward, all models except empty core policy scale with enlarging corridor length. SRMT consistently outperforms baselines both in success rates and in the time needed to solve the task. The shaded area indicates 95% confidence intervals.

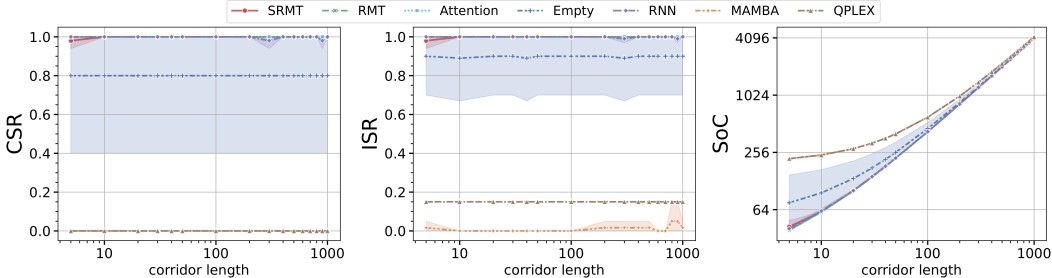

Figure 6: **Directional** reward training leads to all the methods preserving the scores for all tested corridor lengths. The shaded area indicates 95% confidence intervals.

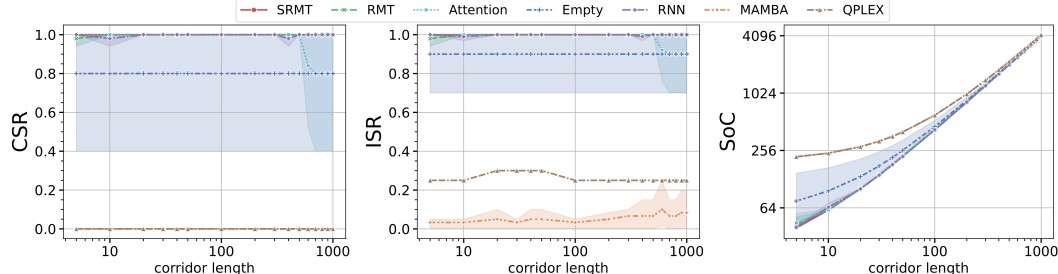

Figure 7: Results of training with **Directional Negative** reward. Vanilla attention fails to scale at corridor lengths of more than 400, compared to the SRMT which preserves the highest scores. That proves the sufficiency of the proposed SRMT architecture. The shaded area indicates 95% confidence intervals.

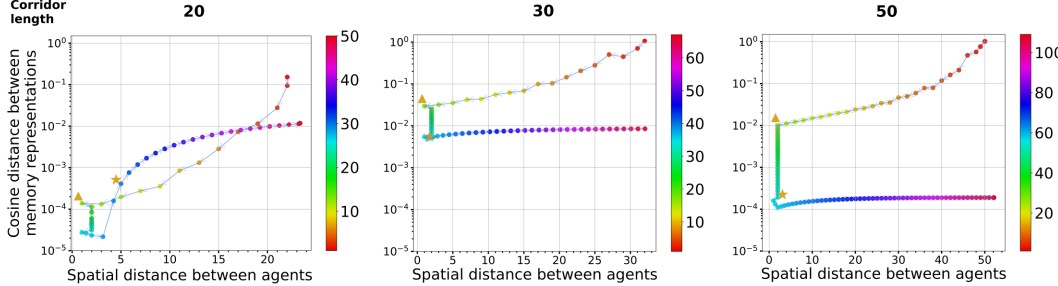

Figure 8: **SRMT memory distances are aligned with the distances between agents.** The figure shows how cosine distances between agents' memory vectors are related to the Euclidean distances between agents on the map for SRMT. The triangle marks the step when agents face each other in the environment, the star shows the episode step when the first goal was achieved. The color bar shows the step number.

We also provide the heatmaps of cosine distances between agent memory states during the episode. Figures 9 and 10 show how SRMT memory for states are related to each other. Figures 11 and 12 depict the paired distances between memory vectors on different time steps in the episode for both agents.

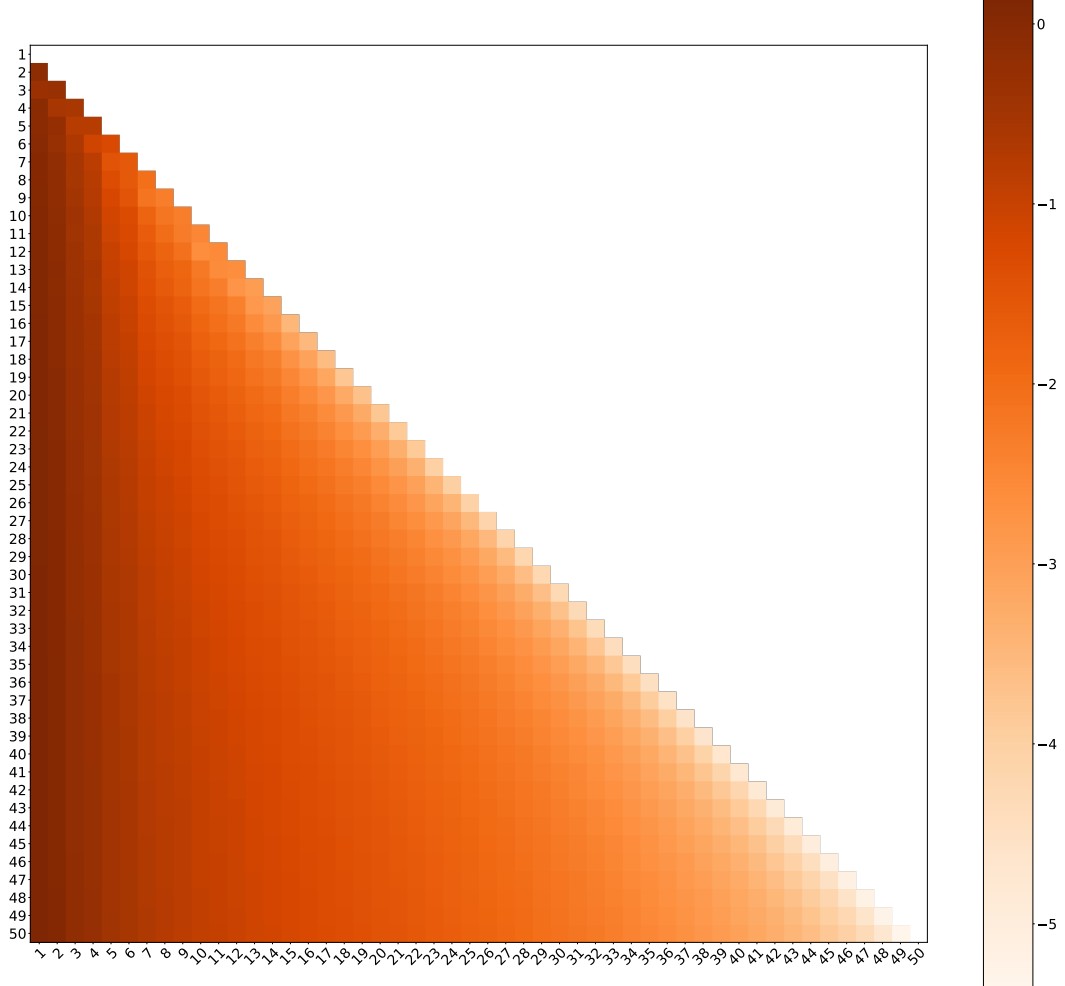

Figure 9: SRMT agent 1 heatmap of distances between memory states.

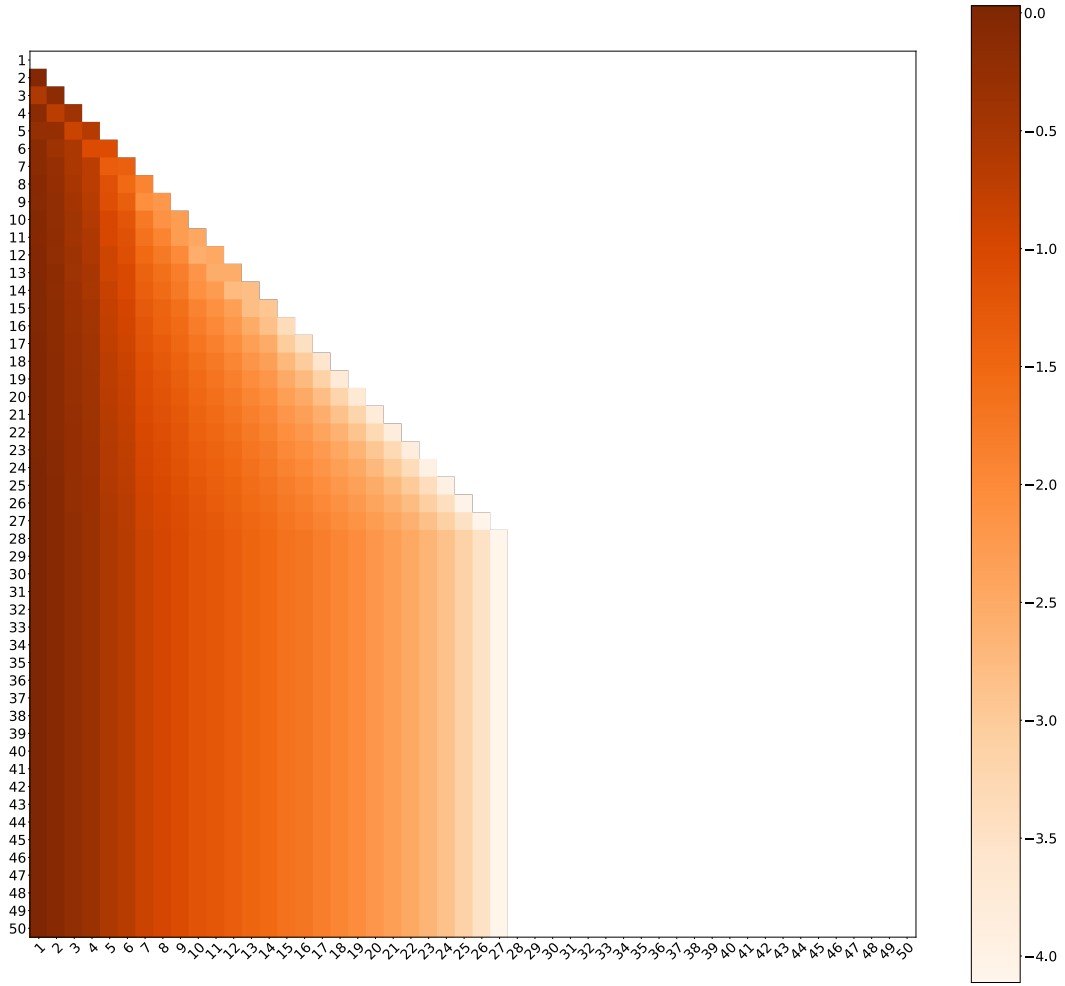

Figure 10: SRMT agent 2 heatmap of distances between memory states.

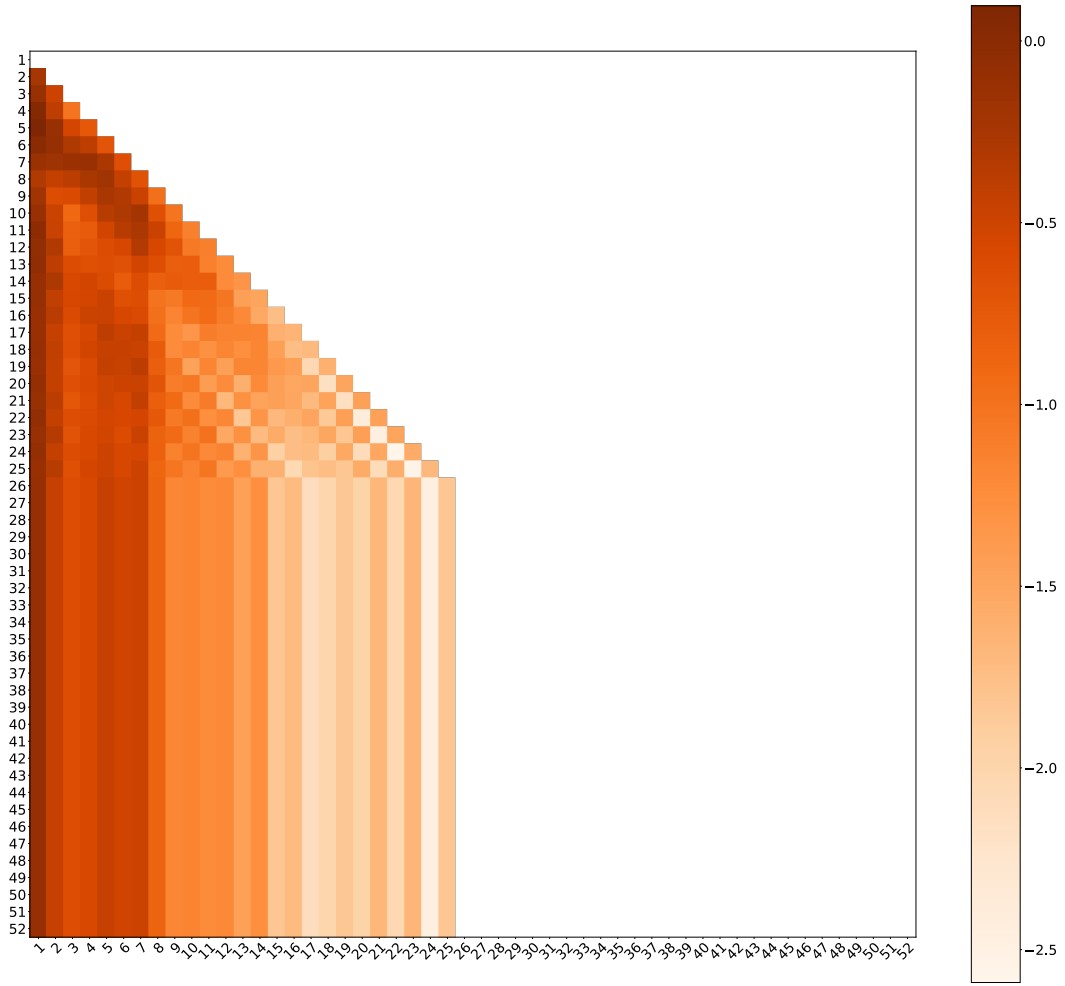

Figure 11: Agent 1 with RMT policy heatmap of distances between memory states.

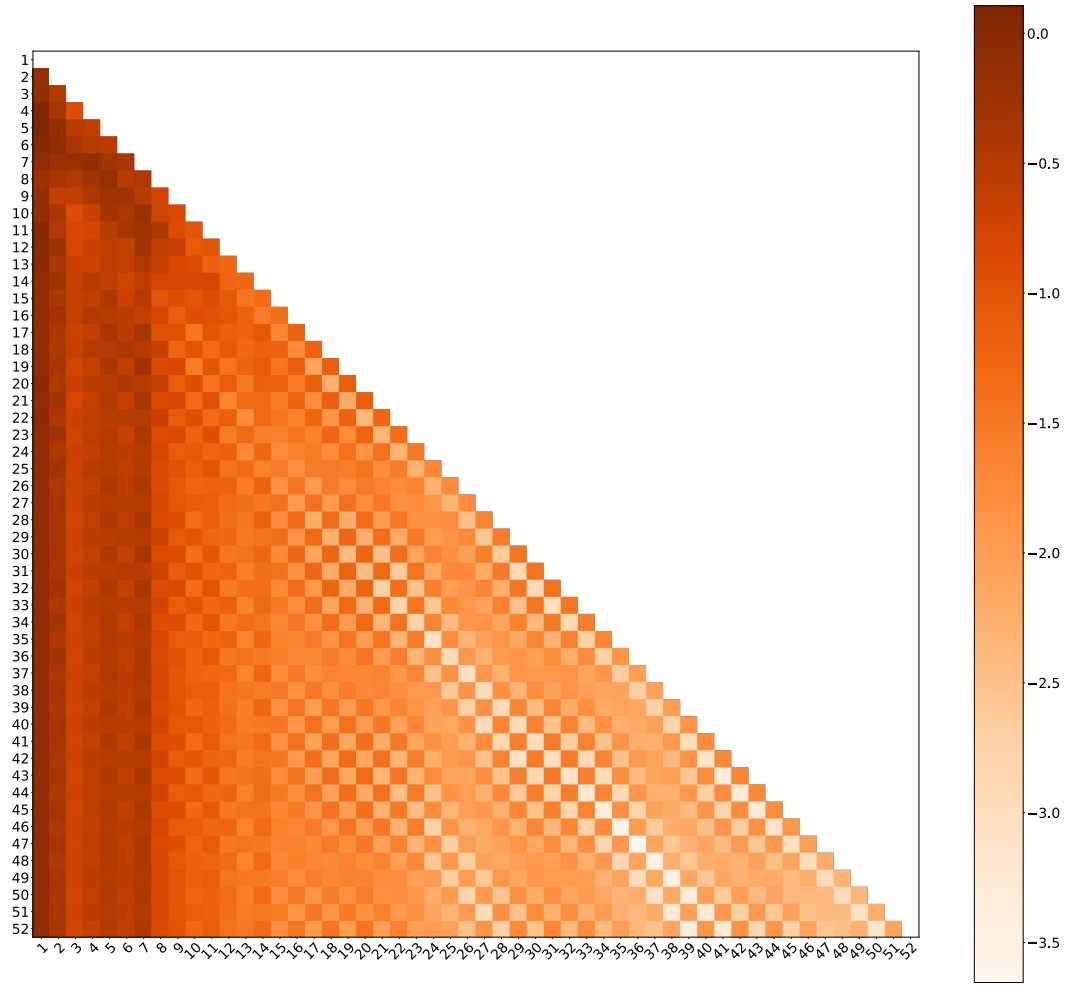

Figure 12: Agent 1 with RMT policy heatmap of distances between memory states.