# OpenReview forum: "Shared Recurrent Memory Improves Multi-agent Pathfinding"
_NeurIPS.cc/2024/Workshop/UniReps — UniReps_

### Official Review · Reviewer_Mjq2 · 2024-09-28
**Review of Shared Recurrent Memory Improves Multi-agent Pathfinding**

**Rating:** 7
**Confidence:** 3

**Review:**

Quality:
The work introduces the Shared Recurrent Memory Transformer (SRMT) as a novel architecture for enhancing coordination in multi-agent system. It demonstrates high quality in terms of the research methodology, experimental design, and presentation of results. The authors have provided detailed information about the proposed Shared Recurrent Memory Transformer (SRMT) and its application in multi-agent systems.

Clarity:
The document is well-structured and clearly presents the proposed SRMT model, the experimental setup, and the results. The use of figures and tables generally aids in understanding the experimental outcomes. One potential improvement could be Figure 3 is a little messy and could be improved to increase clarity.

Originality:
The work introduces a novel approach, the Shared Recurrent Memory Transformer, which extends memory transformers to facilitate information exchange and coordination in multi-agent systems. This original contribution adds value to the field of multi-agent systems and reinforcement learning.

Significance:
The study is significant as it addresses the challenge of coordination in multi-agent systems, a critical area in robotics and artificial intelligence. The results demonstrate the effectiveness of SRMT in enhancing coordination and performance in decentralized multi-agent systems, which has practical implications for real-world applications.

Pros:

1. Introduction of a novel approach, SRMT, to enhance coordination in multi-agent systems.

2. Rigorous evaluation metrics and comparison with baseline models.


Cons:

1. Improve the visualization of experimental result

2. The limitations section could be expanded to provide a more comprehensive understanding of the constraints of the proposed approach.

---

### Official Review · Reviewer_SDJz · 2024-10-04
**MARL with Shared Recurrent Memory Transformer architecture**

**Rating:** 7
**Confidence:** 2

**Review:**

Summary: This paper introduces the Shared Recurrent Memory Transformer (SRMT), a novel architecture for multi-agent coordination in partially observable environments. Inspired by the global workspace theory, SRMT extends memory transformers to multi-agent settings by pooling and broadcasting individual working memories. The authors evaluate SRMT on a bottleneck navigation task, demonstrating improved performance over several baselines, especially under challenging reward structures and when generalizing to longer corridors.

Minor Points:
- The paper provides an abstract but clear motivation for the proposed approach, linking it to established theories in cognitive science.
- The evaluation on different reward structures (directional, moving negative, and sparse) provides good support into the model's performance and robustness.

Major Points:
- Although compared with reasonable baselines, the paper might lack comparisons with other methods in MARL, such as Centralized Critics with Decentralized Actors (CCDA) [1], and without these comparisons, it could be challenging to assess the true contribution of SRMT to the field.

Citation: [1] Lyu, X., Xiao, Y., Daley, B., & Amato, C. (2021). Contrasting centralized and decentralized critics in multi-agent reinforcement learning. arXiv preprint arXiv:2102.04402.

---

### Official Review · Reviewer_zPZo · 2024-10-06
**Sensible extension to RMT for MAS, but weak performance justification**

**Rating:** 5
**Confidence:** 4

**Review:**

Incorporating shared memory in the cross attention block demonstrates the authors have an sound understanding and appreciation for the strengths and weaknesses of a transformer. As such, the paper reads as a natural extension of Recurrent Memory Transformers to multi-agent settings.

The baselines, although well known (Actor-critic,GRU), seem cherry picked.  Justification as to why they picked these baselines would alleviate this concern. Moreover, experiments are run on only one task : POGEMA. Running experiments on at least one other task would  demonstrate the utility of this architecture and alleviate any concerns of cherry picking tasks.

Currently, without the appendix, which the reviewer is not obligated to read, the paper seems a little misleading. For example, the SRMT performance drops below the RNNs for long corridor lengths but nothing is mentioned about it. Overall, the paper is easy to read and understand.

The reviewer recommends that the authors include settings and more baselines along with justifications for the same.

---

### Official Review · Reviewer_UiPg · 2024-10-06
**Sound methodology, SRMT offers minor improvements over RMT with situation-dependent performance, but may diverge from the workshop's focus.**

**Rating:** 7
**Confidence:** 4

**Review:**

This paper introduced a revision for the recurrent memory transformer (RMT). Here this Shared RMT aims to improve the multi-agent coordination.

The methodology is sound, including solid task design for multi-agent coordination, policy learning architecture, nicely designed metrics. The reasons for the model selected are valid though the most proceeding architecture is not implemented. More clarification details will be better, such as the core part is replaced by various benchmarks to assess the SRMT performance.

Based on the results, there are the similarities in performance between SRMT and RMT, and that the improvement of SRMT over RMT may be minor and situation-dependent. In terms of performance, SRMT's results (CSR, ISR, SoC) are largely comparable to the original RMT, particularly in the directional and moving negative reward functions. SRMT may show an advantage in the sparse reward condition, but the statistical significance of this improvement is unclear. Regarding generalization, SRMT outperformed RMT in the extended corridor length scenario for multi-agent pathfinding. However, RMT still demonstrates strong and stable performance, outperforming other baseline models. Given the computing power required, the overall model performance, and the complexity of more realistic pathfinding scenarios (with corridor lengths under 100), RMT might be sufficient for most practical applications.

Additionally, while the paper introduces SRMT to enhance multi-agent coordination, its focus on task-specific performance improvements and pathfinding does not directly address the workshop's core theme of neural representation convergence.

---

### Decision · Program_Chairs · 2024-10-10

**Decision:**

Accept

**Comment:**

In light of the positive reviewers' feedback and relevancy of the submission, we are pleased to accept this paper for presentation at UniReps 2024. We kindly ask the authors to incorporate the reviewers' suggestions and feedback in the final camera-ready version of the manuscript.

---

> ### Author Response · Authors · 2024-11-08
> **Official Comment by Authors**
>
> We would like to sincerely thank all the reviewers for their feedback and positive reviews. We have updated the paper with a few changes:
> - We added the evaluation results for the two additional baseline methods: MAMBA and QPLEX.
> - We specified the statistical significance of the difference in SRMT and RMT scores for the Sparse rewarding function.
> - We added the explanation of the baseline methods choice.
> - We updated the presentation of Figure 3.